# HIV, sexual violence, and termination of pregnancy among adolescent and adult female sex workers in Malawi: A respondent-driven sampling study

Claire Bossard[1]*, Menard Chihana[1], Sarala Nicholas[1], Damian Mauambeta[2], Dina Weinstein[2], Nolwenn Conan[1], Elena Nicco[2], Joel Suzi[3], Lucy OConnell[4], Elisabeth Poulet[1], Tom Ellman[4]

1 Epicentre, Médecins Sans Frontières, Paris, France, 2 OCB, Médecins Sans Frontières, Blantyre, Malawi, 3 The Malawi National Aids Commission, Lilongwe, Malawi, 4 Southern African Medical Unit, Médecins Sans Frontières, Cape Town, South Africa

* claire.bossard@epicentre.msf.org

**Data Availability Statement:** Data are available on request in accordance with MSF's data sharing policy. Requests for access to data should be made

## Abstract

Female Sex Workers (FSWs) are a hard-to-reach and understudied population, especially those who begin selling sex at a young age. In one of the most economically disadvantaged regions in Malawi, a large population of women is engaged in sex work surrounding predominantly male work sites and transport routes. A cross-sectional study in February and April 2019 in Nsanje district used respondent driven sampling (RDS) to recruit women ≥13 years who had sexual intercourse (with someone other than their main partner) in exchange for money or goods in the last 30 days. A standardized questionnaire was filled in; HIV, syphilis, gonorrhea, and chlamydia tests were performed. CD4 count and viral load (VL) testing occurred for persons living with HIV (PLHIV). Among 363 study participants, one-quarter were adolescents 13–19 years (25.9%; n = 85). HIV prevalence was 52.6% [47.3–57.6] and increased with age: from 14.7% (13–19 years) to 87.9% (≥35 years). HIV status awareness was 95.2% [91.3–97.4], ART coverage was 98.8% [95.3–99.7], and VL suppression 83.2% [77.1–88.0], though adolescent FSWs were less likely to be virally suppressed than adults (62.8% vs. 84.4%). Overall syphilis prevalence was 29.7% [25.3–43.5], gonorrhea 9.5% [6.9–12.9], and chlamydia 12.5% [9.3–16.6]. 72.4% had at least one unwanted pregnancy, 17.9% had at least one abortion (40.1% of which were unsafe). Half of participants reported experiencing sexual violence (SV) (47.6% [42.5–52.7]) and more than one-tenth (14.2%) of all respondents experienced SV perpetrated by a police officer. Our findings show high levels of PLHIV-FSWs engaged in all stages of the HIV cascade of care. The prevalence of HIV, other STIs, unwanted pregnancy, unsafe abortion, and sexual violence remains extremely high. Peer-led approaches contributed to levels of ART coverage and HIV status awareness similar to those found in the general district population, despite the challenges and risks faced by FSWs.

to data.sharing@msf.org. For more information please see: 1) MSF's Data Sharing Policy: http://fieldresearch.msf.org/msf/handle/10144/306501 2) MSF's Data Sharing Policy PLOS Medicine article: http://journals.plos.org/plosmedicine/article?id=10.1371/journal.pmed.1001562.

**Funding:** We did not receive an award for the study. The study was entirely funded by Médecins sans Frontières (MSF) Belgium – (Malawi).

**Competing interests:** The authors declare no conflicts of interest.

## Introduction

Female sex workers (FSWs) face high levels of violence, stigma, discrimination, and other human rights abuses [1, 2] that make them extremely vulnerable. They are 14 times more likely to be infected with HIV than other adult women [3, 4], with 17.8% of female HIV infections across sub-Saharan Africa estimated to be attributable to sex work (as much as 41.9% of infections in a South African study) [5, 6]. Younger FSWs have an increased risk of HIV and STI infection, as well as physiological, emotional and social vulnerabilities, common challenges related to stigma, discrimination and criminalization and limited access to services [7–9]. In Malawi, where HIV prevalence remains one of the highest in the world, with variation in gender (10.8% for women; 6.4% for men), age, socio-economic characteristics and geographic location (14.6% for urban areas; 7.4% for rural areas), the rate is five times higher among adult FSWs (62.7%) [10–13]. Nsanje district is one of the most economically disadvantaged areas of Malawi [14] and has a large population of resident and migrant women who sell sex along transport routes at (local and migrant) male labour sites. Women's HIV prevalence among those ≥15 years has been estimated at 14.0% in Nsanje. Nevertheless, progress has been made towards UNAIDS 90-90-90 indicators: after five years of HIV-focused programming by the Ministry of Health (MoH) and Médecins sans Frontières (MSF), 82.4% were said to know their HIV status, 95.7% of PLHIV were on ART, and 90.8% on ART were virally suppressed [15].

Since 2013, MSF has provided HIV, TB, and sexual and reproductive health (SRH) prevention and care services to FSWs in Nsanje using a peer-led, one-stop, sex worker-friendly model [16] following the international recommendations [9]. FSWs are a hard-to-reach and understudied population, and recent district HIV surveys from Nsanje do not provide information on this group despite the fact that their needs are acute and unique, and addressing them will be key to stopping community transmission overall [15]. To date, studies in Malawi have not measured FSWs' awareness of their HIV status using a representative sample. Likewise, younger FSWs have been neglected in research from both Malawi and other global contexts. This leaves critical gaps in our understanding of FSWs' vulnerability to HIV and other risks and the adequacy of services designed to reach them. We recruited a cohort of FSW that included those who began selling sex at a young age (from age 13) to assess whether and how adolescent differ from their adult counterparts. We assessed sexually transmitted infection (STI) prevalence, progress towards UNAIDS 90-90-90 HIV targets, and SRH care behaviours (including unwanted pregnancy, pregnancy termination, and sexual violence) in a region with established HIV and SRH services tailored to the distinct needs of FSWs. Respondent Driven Sampling (RDS) was used to achieve a more robust sample of women in an attempt to better understand this extremely vulnerable group.

## Materials and methods

### Study design, population and setting

We used Respondent Driven Sampling (RDS) to enroll study participants, a technique that weights peer-driven snowball sampling to compensate for its non-random survey design [17]. RDS is widely used to recruit hard-to-reach populations most at-risk of HIV [18, 19]. Three of MSF's five activity sites were purposively selected for the study (Fatima, Bangula and Nsanje Mboma). One site was peri urban (Nsanje Mboma), while the others were more remote and rural. Study eligibility was women ≥13 years, living and/or working in the district for the previous six months, who had sexual intercourse with someone (other than their main partner) in exchange for money or goods in the last 30 days.

## Sampling

Community outreach occurred prior to the study to inform and select "seeds" (i.e. individuals who comprise the first wave of RDS participants) to initiate the process. Per standard RDS methodology and guidance from the Malawian ethical review board (ERB), participants were financially reimbursed for their participation, however all involvement was entirely voluntary. To ensure the diversity of the RDS sample and their networks, seeds were selected to ensure a representative age distribution ($\leq$19; >19 years) and to include those with and without a history of engagement in MSF activities or care. Each seed received an explanation of study objectives and two coded, non-replicable paper recruitment coupons that were used to further enlist their eligible peers into the study. If an eligible peer enrolled in the study, the original seed was reimbursed for their effort and the referred respondent received two recruitment coupons. This continued until the sample size was met. FSWs were requested to recruit known, eligible sex workers (who had not enrolled in the study) from their social networks. In two study sites, five seeds were initially selected, while eight seeds initiated the third study site. Recruitment coupons were coded to provide data on recruiter/recruitee relationships and unique identity information. A maximum of eight "waves" of RDS sampling were performed at each site. Participants were asked three questions to estimate their personal network size. Assuming an HIV prevalence of 50%, an HIV status awareness of 65%, a 95% confidence interval, a precision of 10%, and a design effect of 2 [20], we sought a sample size of 350 FSWs participants.

## Data collection

Participants provided written informed consent to participate in the study. For participants $\leq$ 18 years, assent was obtained from the District Commissioner (DC) who represents the country authorities and provided consent for all minors included in the study. This process which avoids the need to seek parental consent, was approved by the local and international ethics committees. The BBSS survey which took place in a similar context in 2014 also included minors among participant and followed the same process [13]. Indeed, seeking consent from a parent/guardian may have been contrary to the best interest of the young person and may have put them at greater risk (in particular if the adult was not aware of the sex work activity). Also, adolescents aged 13–17 years doing sex work were considered as emancipated minors according to the framework of guidelines for research in the social sciences and humanities in Malawi. Finally, the participation of youths in the study could have been undermined if consent was required from the parent/guardian, due to logistical issues and the need of the parent/guardian to present themself at the study point.

In addition, participants $\leq$18 years wishing to participate but unable to sign the consent form, were given the option of providing verbal consent with a witness signing on her behalf. Consenting women presenting to study sites were screened for eligibility by a trained counselor-educator. When in doubt, the study coordinator determined a prospective participant's eligibility status. A standardized questionnaire included questions about demographics, sex work, lifestyle, contraception, past pregnancies, HIV and STI awareness, access to health care, testing, and prior or current HIV treatment. Interviews were conducted in confidential locations, with clear introductions of confidentiality of data collected in Chichewa (the district's primary language) by the counselor-educator that screened the participant for eligibility. FSWs $\leq$18, survivors of violence, and those who were forced into sex work were referred to relevant social services. Appointments were given (starting at day 4) to control the participant flow and assure privacy during interviews.

## Testing and laboratory procedures

HIV and syphilis rapid tests (Determine Rapid HIV-1/2 Antibody and Syphilis TP) occurred on site. Urine samples were collected for gonorrhea and chlamydia laboratory testing. Tests were conducted using Xpert® CT/NG, a qualitative, real-time Polymerase Chain (RT-PCR) assay for automated detection and differentiation of Chlamydia trachomatis and Neisseria gonorrhea. For PLHIV-FSWs, dried blood samples (DBS) were used for viral load (VL) laboratory tests at Nsanje district hospital. VL was quantified using Abbott Real Time HIV-1 (m2000sp) assay (detection limit: 40 copies/mL), and participants were considered virologically suppressed (VLS) at VL<1,000 copies/ml.

## Other key variables and definitions

We defined "at least one episode of sexual violence" as a participant having been forced to engage in any sexual activity against their will. Inconsistent condom use was defined as a participant not always using a condom when engaging in sexual intercourse with a client. Unsafe abortions were those carried out by the woman herself, by another person at home, or by a traditional healer. To calculate UNAIDS target indicators (90-90-90) for participants living with HIV (PLHIV), we used accepted definitions (i.e. the proportion of participants aware of their positive status; the proportion of those aware who were on ART, the proportion of those on ART who have viral suppression) [21]. Adolescent were those ≤19 years. Analysis based on study population and analysis based on subgroups of study population are presented separately in Table 3.

## Statistical methods

Voltz-Heckathorn's RDS-II estimator was used to produce weighted estimates of population proportions and 95% confidence intervals (CIs) after seed exclusion. As the sampling fraction was unknown, sensitivity analysis comparing the results of the RDS-II estimator against an estimator accounting for population size (successive sampling [SS] estimator) was conducted for different population size estimates. Further, to determine the validity of the RDS-II population estimates, we conducted a series of diagnostic tests as described in Gile et al. [22]. In brief, RDS-II estimates were plotted against the cumulative number of recruits (convergence plot) and per seed (bottleneck plot) for HIV prevalence and for the two criteria used for seed selection (age and enrolment in MSF services). Due to differences noted across sites during diagnostic checks, (differences in recruitment characteristics, varying chain lengths, bottlenecks), site-specific estimates were presented in addition to overall estimates. Difference in RDS estimates across sites were assessed using chi square test and the p-values presented. To examine the difference between adolescents and adults on some key indicators, we conducted logistic regression. These models excluded the seeds, weighted the data by the inverse of reported network size and obtained estimates after adjusting for site. Analyses were conducted using Stata v.16 [23] using the RDS package [24] and sensitivity analysis, diagnostic checks and graphs were conducted using the RDS package in R v.3.4.3 [25, 26].

## Institutional review board approval

Ethics approval was granted from the MSF-ERB (Protocol ID: 1846) and the National Health Sciences Research Committee (NHSRC) in Malawi. All study procedures were in line with the Declaration of Helsinki.

### Patient and public involvement

Consent was obtained directly from patients.

## Results

### Study inclusion

From July to September 2019, 18 seeds recruited 389 participants to join the study. Of these, 26 were ultimately ineligible, leaving a final sample size of 363 women (Table 1). Recruitment chains are presented in the supplementary information.

### Population

Sociodemographic and work characteristics are presented in Table 2. Participants were 13–55 years old (median 26; IQR: 20–34). One-quarter (24.8%) were married and 71.8% had only attended primary school. Nearly all (94.0%) were born in Nsanje district and had always lived there (78.6%). Around one-fifth (17.5%) reported selling sex elsewhere in Malawi prior to doing so in Nsanje district, and 2.8% outside of Malawi. Overall, 47.5% reported having ever enrolled in MSF activities with major differences by site: 83.5% in Fatima, 67.3% in Bangula and 24.8% in Nsanje Boma. Seventy-five percent of women had at least one child, of whom 52.3% reported a first pregnancy before the age of 19.

### Sex work characteristics

The median duration of sex work for all FSWs was 4 years [IQR: 3–7], with 37.8% starting to sell sex before 19 years of age. Half (56.8%) reported 3 or more clients per day. Most FSWs solicited clients in bars and restaurants (84.8%), 47.8% in streets and markets, and 22.3% in beer/bottle shops. Locations used to provide sex were rented rooms (86.5%), homes (53.2%), and outdoors (28.2%). Overall, 86.1% started selling sex because they needed money, 57.9% of which was to support family. Around one-third (32.0%) started selling sex after being abandoned by their husband or family, or after their husband or a family member passed away.

### HIV and STIs

Half of all participants (52.6%; 95% CI 47.3–57.6) were HIV-positive on the day of their enrolment in the study (Table 3). HIV prevalence was much higher in Nsanje Boma (64.1%) than other sites (42.4% and 43.2% respectively, p<0.001) and increased substantially with age, from 14.7% among the youngest women (13–19 years) to nearly all (87.9%) FSWs ≥35 years (Fig 1).

**Table 1. Number of recruits and number of RDS recruitment chain by sites.**

| Site | Number of seeds | Median network size [IQR] | Number of recruits by recruitment chains (and number of waves for each chain) | | | | | | | | Total Inclusions | Total participants without seeds |
|---|---|---|---|---|---|---|---|---|---|---|---|---|
| | | | Chain 1 | Chain 2 | Chain 3 | Chain 4 | Chain 5 | Chain 6 | Chain 7 | Chain 8 | | |
| Fatima | 5 | 7.5 [5–10] | 37 (6) | 8 (3) | 12 (3) | 17 (4) | 11 (3) | | | | 85 | 80 |
| Bangula | 5 | 5 [3–7] | 27 (5) | 23 (4) | 31 (4) | 20 (4) | 25 (5) | | | | 126 | 121 |
| Nsanje Mboma | 8 | 5 [2–10] | 16 (5) | 16 (4) | 19 (6) | 5 (3) | 13 (4) | 34 (5) | 23 (4) | 44 (8) | 170 | 162 |
| Total | 18 | 5 [3–10] | | | | | | | | | 381 | 363 |

**Table 2. Socio demographic and work characteristics of the study population of FSW.**

| | | Fatima (N = 80) | Bangula (N = 121) | Nsanje Mboma (N = 162) | Overall (N = 363) | p-value |
|---|---|---|---|---|---|---|
| | | % (95% CI) | % (95% CI) | % (95% CI) | % (95% CI) | |
| Age group (in years) | 13–19 | 26.0 (17.3–37.1) | 27.4 (19.9–36.3) | 24.7 (18.4–32.2) | 25.9 (21.5–30.8) | 0.09 |
| | 20–24 | 25.3 (16.9–36.0) | 23.6 (16.8–32.1) | 13.3 (8.8–19.5) | 19.4 (15.6–23.8) | |
| | 25–34 | 24.9 (16.8–35.1) | 33.1 (25.5–41.8) | 36.2 (29.3–43.8) | 32.7 (28.1–37.6) | |
| | > = 35 | 23.9 (15.8–34.5) | 15.9 (10.4–23.7) | 25.8 (19.6–33.2) | 22.1 (18.1–26.7) | |
| Marital Status | Married | 32.1 (21.5–42.7) | 29.5 (21.7–38.8) | 18.1 (12.5–25.5) | 24.8 (20.4–30.0) | 0.03 |
| Highest Level of Education Attained | Primary | 68.7 (58.1–77.5) | 76.8 (69.0–83.1) | 69.6 (62.4–75.9) | 71.8 (67.2–76.0) | 0.28 |
| Birth place | Nsanje District | 95.1 (88.8–98.0) | 92.2 (86.7–95.6) | 94.7 (90.6–97.1) | 94.0 (91.4–95.8) | 0.53 |
| Residence | Always lived in Nsanje District | 84.6 (4.8–91.1) | 78.9 (70.6–85.3) | 75.4 (68.1–81.5) | 78.6 (74.0–82.5) | 0.26 |
| District sex work | Nsanje District | 84.4 (74.5–91.0) | 79.3 (71.0–85.7) | 77.7 (70.6–83.6) | 79.7 (75.2–83.6) | 0.55 |
| | Elsewhere in Malawi | 13.3 (7.3–22.9) | 19.2 (13.0–27.4) | 18.3 (12.9–25.2) | 17.5 (13.8–21.9) | |
| | Outside Malawi | 2.3 (0.6–8.8) | 1.5 (0.4–5.9) | 4.0 (1.9–8.2) | 2.8 (1.6–5.0) | |
| Prior enrollment with MSF | Yes | 83.5 (69.9–91.7) | 67.3 (56.5–76.5) | 24.8 (19.4–31.2) | 47.5 (42.1–52.9) | <0.001 |
| | No | 16.5 (8.3–30.1) | 32.7 (23.6–43.5) | 75.2 (68.9–80.6) | 52.5 (47.1–57.9) | |
| Number of children | None | 32.4 (23.0–43.6) | 17.3 (11.4–25.3) | 27.0 (20.6–34.5) | 25.0 (20.7–29.8) | 0.04 |
| | One or more | 67.6 (56.4–77.0) | 82.7 (74.7–88.6) | 73.0 (65.5–79.4) | 75.0 (70.2–79.3) | |
| | | (N = 59) | (N = 106) | (N = 127) | (N = 292) | |
| Age at first pregnancy (in years) | < 19 | 54.4 (41.6–66.8) | 51.5 (42.0–60.9) | 51.9 (43.2–60.5) | 52.3 (46.5–58.0) | 0.93 |
| | ≥ 19 | 45.6 (33.2–58.5) | 48.5 (39.1–58.1) | 48.1 (39.5–56.9) | 47.7 (42.0–53.5) | |
| Duration of SW (in years) | < 3 | 26.6 (18.0–37.4) | 13.4 (8.4–20.8) | 27.5 (21.1–34.9) | 22.6 (18.6–27.2) | 0.01 |
| | ≥ 3 | 73.4 (62.6–82.0) | 86.6 (79.2–91.6) | 72.5 (65.1–78.9) | 77.4 (72.8–81.4) | |
| Age at sex work start (in years) | < 19 | 42.9 (32.5–53.9) | 40.8 (32.4–49.9) | 33.0 (26.2–40.7) | 37.8 (32.9–42.9) | 0.24 |
| | ≥ 19 | 57.2 (46.1–67.6) | 59.2 (50.1–67.6) | 67.0 (59.3–73.8) | 62.2 (57.1–67.1) | |
| Number of clients a day | < 3 | 52.1 (41.1–62.8) | 38.6 (30.2–47.8) | 42.1 (34.6–50.0) | 43.2 (38.1–48.4) | 0.16 |
| | ≥ 3 | 47.9 (37.2–58.9) | 61.4 (52.2–69.8) | 57.9 (50.1–65.4) | 56.8 (51.6–61.9) | |
| Locations used to meet clients | Bars and restaurants | 85.9 (75.2–92.4) | 84.5 (76.0–90.4) | 84.5 (77.4–89.7) | 84.8 (80.3–88.5) | 0.96 |
| | Streets and markets | 40.0 (29.9–51.1) | 40.9 (32.5–49.9) | 56.8 (49.0–64.3) | 47.8 (42.7–523.0) | 0.009 |
| | Beer/bottle shops | 11.6 (6.1–20.8) | 23.7 (16.9–32.2) | 26.5 (20.2–33.9) | 22.3 (18.2–26.9) | 0.03 |

(*Continued*)

**Table 2.** (Continued)

| | | Fatima (N = 80) | Bangula (N = 121) | Nsanje Mboma (N = 162) | Overall (N = 363) | p-value |
|---|---|---|---|---|---|---|
| | | % (95% CI) | % (95% CI) | % (95% CI) | % (95% CI) | |
| Locations used to provide sex | *Rented rooms* | 85.4 (75.5–91.7) | 87.7 (80.2–92.6) | 86.2 (79.8–90.8) | 86.5 (82.4–89.7) | 0.89 |
| | *Homes* | 33.8 (24.4–44.8) | 55.7 (46.7–64.4) | 61.1 (53.3–68.4) | 53.2 (48.1–58.4) | <0.001 |
| | *Outdoors* | 28.2 (19.6–38.9) | 22.4 (16.0–30.6) | 32.6 (25.9–40.2) | 28.3 (23.9–33.0) | 0.16 |
| Reasons for selling sex | *Needed money* | 75.2 (66.0–82.6) | 88.6 (83.4–92.3) | 88.9 (84.6–92.1) | 86.1 (83.0–88.7) | 0.001 |
| | *Support family* | 39.8 (29.4–51.1) | 64.6 (55.9–72.5) | 61.6 (54.0–68.7) | 57.9 (52.8–62.9) | 0.001 |
| | *Abandonment or death of husband or family member* | 34.1 (24.7–44.9) | 33.0 (25.4–41.7) | 30.1 (23.7–37.5) | 32.0 (27.5–36.9) | 0.78 |

% (95% CI) weighted RDS estimates, N = denominator

In terms of STIs, almost a third (29.7%) of participants were positive for syphilis, 12.5% for chlamydia, and 9.5% for gonorrhoea.

Among PLHIV-FSWs, nearly all (95.2%) reported being aware of their HIV-positive status, of whom 98.8% were taking ART. Of those who took ART, 83.2% were virologically supressed. For these indicators, no differences were observed across site.

## HIV prevention & knowledge

All but one study participant knew at least one HIV prevention method (N = 362). Condom use was cited by all participants who knew at least one. Taking ARV before sexual intercourse (pre-exposure prophylaxis [PreP]) and abstinence were cited as prevention methods by only 2.2% and 2.1% of participants, respectively. Among the FSWs who knew how to prevent HIV after unprotected sex (N = 196), nearly all (89.1%) cited post-exposure prophylaxis (PEP). Nearly all (89.0%) HIV-negative participants felt that they had a "moderate" or "high" risk of acquiring HIV (158/176). However, among the HIV-negative FSWs who knew of PEP, less than half (41.1%) reported having ever used it. Fifty-three percent of all FSWs inconsistently used condoms with clients. Among FSWs who reported irregular condom use (N = 136), 60.0% did not use a condom with their last client. Our data also showed that more HIV-positive FSWs reported inconsistent use of condoms than the HIV-negative (59.8% vs. 45.6%, p = 0.0075, results not presented in table).

## Contraceptives, termination of pregnancy, sexual violence

In total, 83.2% reported using contraceptives, though this number dropped to 60.8% when condoms were excluded as an option. Injectable contraceptives were most common, used by 65.8% of those who used methods other than condoms, followed by implants (27.1%) (results not presented in table). Three-quarters (72.4%) reported having had at least one unwanted pregnancy and 17.9% reported having had at least one abortion, nearly half (40.1%) of which were unsafe and used plants or tablets from the market. Nearly half of participants reported at least one episode of sexual violence (47.6%). More than half (56.8%) of these women had experienced SV in the last month. Rape by a police officer was reported in 14.2% of FSWs who had ever experienced sexual abuse.

**Table 3. Weighted RDS estimates of key study indicators.**

| Population/Indicator | Fatima | Bangula | Nsanje Mboma | Overall | P-value |
|---|---|---|---|---|---|
| | % (95% CI) | % (95% CI) | % (95% CI) | % (95% CI) | |
| **Analysis based on study population** | | | | | |
| **Among FSWs** | N = 79–80 | N = 121 | N = 161–162 | N = 361–363 | |
| HIV Prevalence | 42.4 (32.0–53.5) | 43.3 (34.7–52.4) | 64.1 (56.4–71.1) | 52.6 (47.3–57.6) | <0.001 |
| STI Prevalence | | | | | |
| *Syphilis* | 28.5 (19.9–39.1) | 31.8 (24.2–40.4) | 28.7 (25.3–34.5) | 29.7 (25.3–43.5) | 0.82 |
| *Chlamydia* | 12.9 (6.9–22.9) | 10.3 (5.8–17.7) | 14.0 (9.3–16.6) | 12.5 (9.3–16.6) | 0.68 |
| *Gonorrhea* | 7.0 (3.2–14.8) | 9.2 (5.3–15.6) | 11.0 (7.1–16.6) | 9.5 (6.9–12.9) | 0.60 |
| Knowledge of methods to protect against HIV infection | | | | | |
| *Condom use* | 100 | 100 | 100 | 100 | |
| *Taking ARV before sexual intercourse* | 3.7 (1.5–8.8) | 2.5 (1.9–5.8) | 1.1 (0.4–3.4) | 2.2 (1.2–3.7) | 0.21 |
| *Abstinence* | 0 | 3.8 (1.2–11.1) | 1.9 (0.5–7.4) | 2.1 (0.9–5.0) | 0.32 |
| Knowledge of methods to use in case of unprotected sex | 53.5 (42.4–64.2) | 71.1 (62.1–78.8) | 33.9 (27.1–41.4) | 50.2 (45.0–55.4) | <0.001 |
| Inconsistent use of condom with clients | 44.4 (33.7–55.6) | 45.8 (37.0–54.8) | 62.3 (54.7–69.4) | 53.0 (47.8–58.1) | 0.005 |
| Contraceptives use | 80.9 (71.2–87.9) | 87.5 (80.6–92.1) | 81.1 (74.6–86.3) | 83.2 (79.1–86.6) | 0.27 |
| Contraceptive type | | | | | |
| *Condoms* | 18.1 (10.3–29.8) | 19.9 (13.2–28.9) | 31.4 (23.8–40.0) | 24.5 (19.9–29.8) | 0.05 |
| *Injectable* | 57.7 (45.2–69.3) | 48.9 (39.3–58.5) | 51.6 (42.9–60.2) | 51.9 (46.2–57.7) | |
| *Implants* | 20.3 (12.1–32.0) | 28.3 (20.5–37.7) | 13.7 (8.8–20.8) | 20.3 (16.1–25.2) | |
| *Other* | 3.9 (1.4–10.0) | 2.9 (1.2–6.9) | 0.3 (1.6–6.9) | 3.3 (2.0–5.4) | |
| Unwanted pregnancy | 76.7 (64.3–85.7) | 73.1 (63.9–80.7) | 69.8 (61.3–77.1) | 72.4 (67.0–77.2) | 0.60 |
| Ever experienced sexual violence | 50.1 (39.2–60.9) | 49.2 (40.4–58.1) | 45.1 (37.6–52.7) | 47.6 (42.5–52.7) | 0.70 |
| **Analysis based on subgroups of study population** | | | | | |
| **Among HIV positive FSWs (PLHIV-FSWs)** | N = 30–33 | N = 47–51 | N = 96–102 | N = 173–186 | |
| HIV Cascade Indicators | | | | | |
| *Aware of HIV-positive status* | 94.6 (80.5–98.7) | 96.5 (86.9–99.1) | 94.8 (88.7–97.7) | 95.2 (91.3–97.4) | 0.86 |
| *On ART* | 100 | 95.7 (84.2–98.9) | 100 | 98.8 (95.3–99.7) | 0.07 |
| *VL Suppression* | 91.0 (75.2–97.1) | 80.6 (67.3–89.4) | 82.0 (73.3–88.3) | 83.2 (77.1–88.0) | 0.41 |
| **Among FSWs who knew methods to prevent HIV** | N = 45 | N = 90 | N = 61 | N = 196 | |
| Cited PEP | 88.1 (72.5–95.4) | 95.5 (86.8–98.5) | 80.7 (67.2–89.5) | 89.1 (82.9–93.2) | 0.04 |
| **Among HIV negative FSWs who knew methods to use in case of unprotected sex** | N = 28 | N = 50 | N = 18 | N = 96 | |
| Ever used PEP | 53.8 (35.3–71.2) | 31.1 (19.4–45.9) | 48.2 (26.5–70.6) | 41.1 (31.4–51.5) | 0.13 |
| **Among HIV negative FSWs** | N = 46 | N = 70 | N = 60 | N = 176 | |
| Self-estimated of risk of contracting HIV / AIDS | | | | | |
| *No or low risk* | 18.6 (9.6–33.1) | 6.2 (2.3–15.4) | 10.8 (4.9–22.0) | 11.0 (6.2–15.7) | 0.13 |
| *Moderate to high risk* | 81.4 (67.0–90.4) | 93.8 (84.6–97.7) | 89.2 (78.0–95.1) | 89.0 (83.2–93.0) | |
| **Among FSWs with inconsistent use of condoms** | N = 0 | N = 42 | N = 94 | N = 136 | |
| Did not use a condom last time | N/A* | 59.6 (43.7–73.6) | 60.2 (49.6–69.9) | 60.0 (51.2–68.2) | 0.95 |
| **Among FSWs who already experienced sexual violence** | N = 43 | N = 64 | N = 79 | N = 186 | |
| Experienced sexual violence last month | 72.1 (58.0–82.8) | 57.4 (45.1–68.9) | 47.0 (35.9–58.5) | 56.8 (49.6–63.7) | 0.02 |
| Ever raped by a police officer | 17.0 (8.6–30.8) | 11.4 (5.7–21.3) | 15.0 (8.8–24.4) | 14.2 (10.0–19.8) | 0.67 |

[a]No data were collected for site 1 for the question "*Among inconsistent use, did not used a condom last time?*"

% (95% CI) weighted RDS estimates, N = denominator

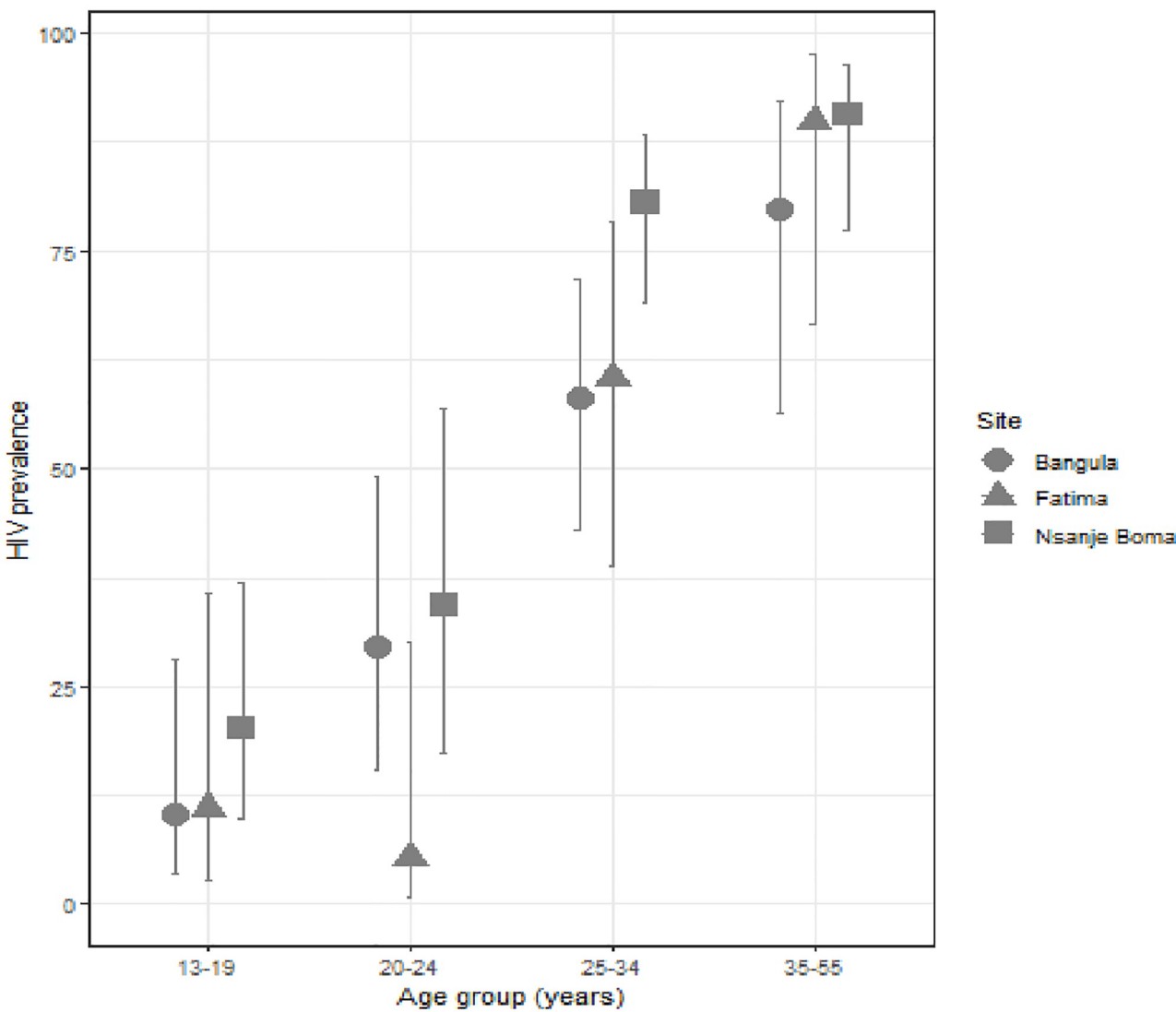

**Fig 1. HIV Prevalence (%) by age group and study sites.** Overall prevalence by age group: 13–19 years:14,7%; 20–24: 24.2%; 25–34: 69.5%; 35–55: 87.9%.

## Adolescents

One-quarter of participants were adolescents 13–19 years (25.9%, Table 4). Among them, 7 were ≤16 years of age (none of these individuals were HIV positive). The overall HIV prevalence among adolescents was 14.7% (Fig 1). Adolescents were three times as likely to report having had at least one unwanted pregnancy than adults (88.6% vs. 69.6%, OR = 3.5, p = 0.025), though they reported similar level of sexual violence as older FSWs (49.3% vs. 47.0%, OR = 1.09, p = 0.74). UNAIDS 90-90-90 target indicators for adolescents were lower than adults (adolescents: 85.0, 100.0, 62.8 vs. adults: 95.9, 98.7, 84.4).

## Discussion

This study is the first to use an RDS methodology to research vulnerable, hard-to-reach FSWs in Malawi, a population that has been systematically overlooked in much of the HIV research

**Table 4. Logistic regression for adolescents (13–19 years) compared to adults (≥ 20 years) on some key indicators.**

| | | Weighted RDS estimates | | | | OR (95% CI) | P-value |
|---|---|---|---|---|---|---|---|
| | | Adolescents (13–19 years) | | Adults (≥ 20 years) | | | |
| | | N | % (95% CI) | N | % (95% CI) | | |
| HIV Prevalence | | 85 | 14.7 (8.5–24.2) | 278 | 63.7 (57.9–69.2) | 0.09 (0.05–0.17) | <0.001 |
| Among at least one pregnancy, at least one unwanted pregnancy | | 35 | 88.6[a] (73.3–96.8) | 257 | 69.6 (63.6–75.2) | 3.45 (1.17–10.17) | 0.025 |
| Ever Experienced sexual violence | | 85 | 49.3 (38.8–59.8) | 278 | 47.0 (41.2–52.9) | 1.09 (0.67–1.77) | 0.74 |
| Among HIV positive HIV Cascade of care (90-90-90) | *Aware of HIV-positive status* | 12 | 85.0 (55.0–96.3) | 174 | 95.9 (92.0–98.0) | 0.24 (0.05–1.30 | 0.10 |
| | *On ART* | 10 | 100 | 166 | 98.7 (95.0–99.7) | 1 | |
| | *VLS* | 10 | 62.8 (32.0–85.8) | 163 | 84.4 (78.3–89.1) | 0.30 (0.08–1.15) | 0.08 |

[a]non weighted RDS adjusted estimate

in the country to date. It is also one of the few HIV studies in Africa to actively recruit adolescent who sell sex into the sample. Encouragingly, PLHIV-FSWs were aware of their HIV status, were on ART, and many were virally suppressed, in some cases surpassing the UNAIDS 90-90-90 targets (our sample was at 95.2%, 98.8%, 83.2%, similar to the general female population of the district) [15]. Yet, these positive notes become more sombre in light of the high prevalence of HIV among some FSWs, with nearly all of those >35 having acquired the disease during their lives (with high STI levels added to risk of contracting and transmitting HIV) [27–29].

Moreover, the study highlighted how extremely vulnerable women who sell sex are in the district. Similar to other studies from the region, our sample was largely comprised of young, single mothers [27, 30–32]. They engaged in frequent sex work, often as the sole source of income, and were regularly exposed to sexual violence, sometimes perpetrated by the police. The nearly half who experienced sexual violence (47.6%) may be substantially underestimated due to denial, trauma, or lack of recognition of sexual abuse. Also, underreporting and recall bias are likely to affect estimates. For example, while only half of FSWs reported ever having experienced SV, the fact that so many of these (56.8%) reported it in the last month suggest the true lifetime experience of SV may be far higher [33]. Most FSWs knew about HIV transmission and prevention and the risks inherent to their profession, yet more than half used condoms irregularly (53.0%), underlining the difficulties of negotiating condom use with clients [34] and better pay for unprotected sex [35]. Inconsistent condom use was even greater for FSWs living with HIV, some of whom were actively viremic and therefore more likely to transmit disease. For these women, prevention measures must be more readily accessible if this number is to improve [36].

WHO HIV target indicators were relatively similar for FSWs and the general female population: awareness of HIV status was higher among FSWs (95.2% vs. 82.4%, p = 0.004), ART coverage was similar (98.8% vs. 95.7%, p = 0.055), and VL suppression rates were lower (83.2% vs. 90.8% p = 0.023). Yet, interpretation of these numbers is complex. While FSWs have higher risks and higher needs, they also face many barriers to care [37]. The high ART coverage, after several years of FSW-targeted programming, demonstrates that excellent outcomes can be

achieved through a combination of community, peer-led, approachable, and responsive facility-based services [38]. Nonetheless, lower levels of viral suppression in this group highlights the need for continued investment in these services and increased adherence support. HIV status awareness and VL levels were particularly low among adolescent FSWs, similar to other settings [7], highlighting the need for tailored approaches. Women receiving services from MSF-supported facilities were less likely to be virally suppressed (92.4% vs. 71.9%, p = 0.008), suggesting that the peer-led approach may be identifying those FSWs with the greatest need for support.

Over half of study participants (52.3%) had their first pregnancy ≤19 years, nearly double the rate of the general population (29% of women age 15–19) [12], and four times greater than that reported in a similar South African cohort (14%) [33]. Unwanted pregnancies were extremely common in this group (72.4% reported at least one). The many young, single mothers and the frequent reports of pregnancy termination (despite the fact that abortion is illegal in Malawi) [39] highlight how knowledge of and access to quality contraception choices remains critical, especially long-acting methods, targeting adolescent FSWs [40]. For FSWs, comprehensive SRH must go beyond condom distribution to include access to emergency contraceptives and safe abortion care (SAC). It is also important to enhance the skills of health care providers to address unmet SRH needs among FSWs, including through specific 'values clarification' workshops to address beliefs and attitudes towards sex workers [2, 41].

Adolescents who sell sex merit special attention since they may not have the ability to negotiate condom use and are even more vulnerable to stigma, violence, and discrimination [3, 42–44], which may further reduce their ability to seek support and health services. Similar to findings from other studies, adolescent study participants in this cohort had similar levels of HIV and STI risk factors and exposure to SV as adults, but less understanding of HIV and lower results of viral suppression [7]. Medical needs are pressing in this population and targeted, comprehensive services, including psycho-social care, are gravely needed [43, 45].

The role of peer-led interventions is key for the FSW community [16, 46]. Sex-worker peers should help design the services that will serve their unique needs. Peers can also offer alternative routes to care, prevention, legal, and social support services [47], though improving staff attitudes and friendliness at health facilities is also important. For adolescents who may not respond as well to adult peer-led programming, adolescent peers should be included in programming [48].

Several PLHIV-FSWs who were unaware of their HIV infection or not linked to care were identified and referred to MSF during the RDS study. With its reliance on social networks, RDS can access populations who avoid public venues and are excluded by general population surveys, since peers recruit hidden populations with more ease than outreach workers and researchers [49]. Because of its relatively simple implementation, the use of this method into programmatic activities could be valuable.

Our study is limited by its cross-sectional design, providing a snapshot of a specific period but preventing us from conducting outcome analysis. The study is also dependent on the self-reported data of participants. Recruitment occurred in three geographically distinct sites, and though site specific RDS-estimates fulfill theoretical RDS assumptions, district-level RDS-estimates may potentially be biased because we assume that participants from across this district are well-connected and that social networks may be linked (we did not collect sufficient information on the extent of an individual's network between sites). Further, diagnostic plots revealed short chains and distinctive bottlenecks, especially among younger FSWs in some sites (S2 Fig). This resulted in different HIV prevalence estimates across sites (S1 Fig). Since no other actors working with FSWs were identified in the area, we were restricted to choosing seeds among those engaged in the MSF network. This may have biased our sample if those

linked to MSF were poorly networked with other groups. It was also difficult to recruit adolescent sex workers as initial seeds, since they tended to self-identify as FSWs less frequently than adults [43]. There is a frequent misuse of the penal code by police in Malawi (where sex work is not penalized), and adolescents selling sex are often criminalized and arrested, discouraging them from coming forward for support and health services [39].

## Conclusion

Selling sex carries an enormous amount of risk for the (often very young women) who do so. HIV, sexual violence, unplanned pregnancy, and unsafe abortion are all regular features of their challenging work. In this remote and rural population, investment in friendly, facility-based services and peer-led outreach activities provided a comprehensive package of HIV prevention and care services and allowed FSWs to achieve similar outcomes along the HIV cascade as women in the general population. This occurred despite the fact that FSWs are typically a difficult-to-reach, high risk, and vulnerable group.

Nonetheless, persistent HIV transmission risk remains a concern in this population. Increased investment in facility-led, friendly services as well as synergistic, community-based, peer-led activities is essential if these women's needs are to be met. Structural interventions are also needed, including viable and reliable referral pathways to legal solutions and justice for sexual violence survivors. Awareness raising for stakeholders with a duty to protect these women (such as the police, brothel owners, bar tenants, and clients) is also necessary to reduce violence and stigma and allow FSWs better access to services. Sex workers' voices must also be heard, not only as recipients of care but as information-rich counterparts when adapting services to fit their needs.

## Supporting information

**S1 Fig. Chains of recruitment in the 3 sites.** Each dot represents a participant in the study and the lines represent the passing of recruitment coupons. Blue dots represent women who tested HIV positive and orange circles HIV negative. The dots at the top of each tree represent the seeds.
(TIF)

**S2 Fig. Diagnosis analysis.** a/HIV prevalence; b/Age group; c/Ever enrolled in MSF activities.
(PDF)

## Acknowledgments

The authors are grateful to all study participants in the three study sites in Nsanje district, as well as their children who often accompanied them. We would like to thank the study team: Damian Mauambeta, Charity Mughogho, Linly Chinyama, Debra Sambo, Constance Mwenechanya, Gilbert Mwandira, Richard Jamester, Margaret Kasonya, Christina Kalulu, Daniel Chavuta and Mr Shaba, as well as all staff involved in MSF teams who helped with the preparation and implementation of the study. We would like to thank the traditional authorities of Fatima, Bangula and Nsanje Boma as well as the staff at Nsanje District Hospital laboratory.

## Author Contributions

**Conceptualization:** Claire Bossard, Sarala Nicholas, Nolwenn Conan, Elena Nicco, Lucy OConnell, Elisabeth Poulet, Tom Ellman.

**Data curation:** Menard Chihana.

**Formal analysis:** Claire Bossard, Menard Chihana, Sarala Nicholas.

**Investigation:** Claire Bossard, Damian Mauambeta, Dina Weinstein.

**Methodology:** Claire Bossard, Sarala Nicholas, Nolwenn Conan, Elena Nicco, Joel Suzi, Lucy OConnell, Elisabeth Poulet, Tom Ellman.

**Project administration:** Dina Weinstein, Elena Nicco.

**Resources:** Dina Weinstein, Elena Nicco, Elisabeth Poulet.

**Supervision:** Claire Bossard, Damian Mauambeta, Elisabeth Poulet.

**Validation:** Claire Bossard, Sarala Nicholas.

**Visualization:** Sarala Nicholas.

**Writing – original draft:** Claire Bossard, Sarala Nicholas, Elena Nicco, Tom Ellman.

**Writing – review & editing:** Menard Chihana, Damian Mauambeta, Dina Weinstein, Nolwenn Conan, Joel Suzi, Lucy OConnell, Elisabeth Poulet.

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
