## [Decision Letter · Decision Letter 0]

6 Jul 2022

PONE-D-22-01162HIV, sexual violence, and termination of pregnancy among adolescent and adult female sex workers in Malawi: a respondent-driven sampling studyPLOS ONE

Dear Dr. BOSSARD,

Thank you for submitting your manuscript to PLOS ONE. After careful consideration, we feel that it has merit but does not fully meet PLOS ONE’s publication criteria as it currently stands. Therefore, we invite you to submit a revised version of the manuscript that addresses the points raised during the review process.

We look forward to receiving your revised manuscript.

Kind regards,

Henry F. Raymond

Academic Editor

PLOS ONE

Journal Requirements:

a) Did participants provide their written or verbal informed consent to participate in this study?

b) If consent was verbal, please explain i) why written consent was not obtained, ii) how you documented participant consent, and iii) whether the ethics committees/IRB approved this consent procedure."

3. You indicated that you had ethical approval for your study. In your Methods section, please ensure you have also stated whether you obtained consent from parents or guardians of the minors included in the study or whether the research ethics committee or IRB specifically waived the need for their consent.

6. Please include a copy of Table 3 which you refer to in your text on page 9.

Additional Editor Comments (if provided):

Dear Authors,

This is a well received manuscript. I would like you to address the comments of all reviewers before rendering a final decision.

Thank you for your work.

Reviewers' comments:

Reviewer's Responses to Questions

**Comments to the Author**

1. Is the manuscript technically sound, and do the data support the conclusions?

Reviewer #1: Yes

Reviewer #2: Partly

Reviewer #3: Yes

2. Has the statistical analysis been performed appropriately and rigorously? 

Reviewer #1: Yes

Reviewer #2: Yes

Reviewer #3: Yes

3. Have the authors made all data underlying the findings in their manuscript fully available?

Reviewer #1: Yes

Reviewer #2: Yes

Reviewer #3: Yes

4. Is the manuscript presented in an intelligible fashion and written in standard English?

Reviewer #1: Yes

Reviewer #2: No

Reviewer #3: Yes

5. Review Comments to the Author

Reviewer #1: 1. Giving information about urban and rural variations observed in HIV prevalence and other sexual behaviours indicators will be helpful for programme implementation by The Malawi National AIDS Commission, Malawi.

2. In general, the 15-24 age group HIV positive case is considered as a proxy for incidence since the study focussed only on younger FSWs. Removing the known HIV positives from databases and calculating the incidence % may be helpful to the programme.

3. In table 2, row (locations for providing SW) under the Nsanje Mboma site, N needs to be replaced by the actual number.

Reviewer #2: General Comments: This paper estimated HIV prevalence, HIV Cascade of Care, and STIs Prevalence among female sex workers (FSWs) in Malawi and by using 363 RDS samples. Moreover, this paper also examined FSWs' risk behaviors and sexual violence experience. Authors used Respondent-Driven Sampling (RDS) to recruit participants for a cross-sectional survey, which is appropriate given the ability of RDS to generate statistically unbiased estimates under certain conditions. Authors did clearly present methods of data collection, RDS estimators they used and sensitivity analysis they conducted. However, results of this paper, including tables, figures and interpretations, are somewhat flawed.

Major Comments:

1. Statistical tests were not mentioned in methods section or presented in tables/figures. However, authors used p-values in results section and discussion section.

2. Although author provided sub-headers in results section, which may help readers to easily understand, contents under each sub-header are still disorganized. Many results cannot be traced back to or supported by tables and figures. For instance, odds ratios that author mentioned in Line 155 and Line 156 are not found in tables or figures; VL and CD4 in Line 176 are not found in tables or figures. Results section should be more concise. Authors are highly recommended to keep comprehensive information in tables or figures and only interpret main findings. Authors may also need to tweak their interpretations, such as accurately interpreting odds ratios and avoiding using rates because of the study design.

3. Most of percentages or proportions in results section and discussion section are somewhat confusing. Authors may need to clearly present numerators and denominators for all percentages/proportions.

4. Table 2 and Table 3 are difficult to read. In the last column of Table 2, unexplained values do exist. Some values are confusing because authors did not present stratum-specific counts, numerators, and denominators. For instance, what does the number of being aware of HIV-positive status in Table 3 truly represent?

Minor Comments:

1. In Line 37, what is the reference for the rate among adult FSWs to be five times higher?

2. In Line 169, “95% CI” may be appropriate.

3. In Figure 1, authors are highly recommended to present overall HIV prevalence by age and site-specific HIV prevalence by age in single one plot.

4. Figure 2 looks unnecessary.

5. In Line 222, two percentages have completely different denominators, thus they should not be directly compared.

6. In Line 272, authors said site specific RDS-estimates fulfill theoretical RDS assumptions. However, bottleneck plots in supplementary document may indicate preferential recruitment patterns and homophily. For instance, FSWs with a history of being enrolled in MSF activities were oversampled. Is it possible that FSWs with a history of being enrolled in MSF activities had higher HIV awareness and better treatment compliance and viral suppression comparing to FSWs without a history?

Reviewer #3: This paper describes an RDS study of a population of FSW in Malawi. These women are very vulnerable, and the inclusion of adolescent participants makes this a particularly important dataset.

I was impressed with the thoughtfulness and care that the authors demonstrated with this study. The inclusion of diagnostic plots as well as a frank discussion of convergence and bottleneck issues was quite refreshing. Many papers simply brush these issues under the rug.

I am comfortable recommending the paper for publication as is. Below are some optional comments for improvement.

* I am not an expert on appropriate nomenclature; however, the authors may wish to review the application of the category “sex worker” to minors. The term implies a level of consent that may not be applicable to a trafficked 14 year old.

* Diagnostic results are presented in the discussion section, whereas it may be more appropriate to put this in the results section.

* It is probably worth noting that bottlenecks are to be expected between sites as they are geographically distinct and RDS chains cannot cross from one site to another. Even if the underlying network is well connected between sites, the RDS assumptions are violated because the chains cannot cross between them.

* In addition to citing Stata and R, I recommend citing the RDS specific Stata and R packages used.

* Line 78: typo “two oded”

6. PLOS authors have the option to publish the peer review history of their article (what does this mean?). If published, this will include your full peer review and any attached files.

Reviewer #1: **Yes: **Santhakumar Aridoss

Reviewer #2: No

Reviewer #3: No

---

## [Author Response · Author response to Decision Letter 0]

3 Nov 2022

General comments:

Thank you for your comment. We did edit the article to meet PLOS ONE's style requirements. 

a) Did participants provide their written or verbal informed consent to participate in this study?

b) If consent was verbal, please explain i) why written consent was not obtained, ii) how you documented participant consent, and iii) whether the ethics committees/IRB approved this consent procedure."

We added these details to the ethic statement: 

Participants provided written informed consent to participate in the study. For participants ≤ 18 years, assent was obtained from the District Commissioner (DC) who represents the country authorities and provided consent for all minors included in the study. This process which avoids the need to seek parental consent, was approved by the local and international ethics committees. The BBSS survey which took place in a similar context in 2014 also included minors among participant and followed the same process (Malawi Biological and Behavioural Surveillance Survey Report, 2014). Indeed, seeking consent from a parent/guardian may have been contrary to the best interest of the young person and may have put them at greater risk (in particular if the adult was not aware of the sex work activity). Also, adolescents aged 13-17 years doing sex work were considered as emancipated minors according to the framework of guidelines for research in the social sciences and humanities in Malawi. Finally, the participation of youths in the study could have been undermined if consent was required from the parent/guardian, due to logistical issues and the need of the parent/guardian to present themself at the study point. In addition, women ≤18 years wishing to participate but unable to sign the consent from, were given the option of providing verbal consent with a witness signing on her behalf. 

3. You indicated that you had ethical approval for your study. In your Methods section, please ensure you have also stated whether you obtained consent from parents or guardians of the minors included in the study or whether the research ethics committee or IRB specifically waived the need for their consent.

Thank you for your comment. This paragraph was added to the methods section: 

Participants provided written informed consent to participate in the study. For participants ≤ 18 years, assent was obtained from the District Commissioner (DC) who represents the country authorities and provided consent for all minors included in the study. This process which avoids the need to seek parental consent, was approved by the local and international ethics committees. In addition, women ≤18 years wishing to participate but unable to sign the consent from, were given the option of providing verbal consent with a witness signing on her behalf

We did not receive an award for the study. The study was entirely funded by Médecins sans Frontières (MSF) Belgium – (Malawi). This sentence was added in the funding information section. 

Thank you for your comment. We added a caption to figure 1 (page 13) and opted to deleted figure 2, as recommended by reviewer 2. 

6. Please include a copy of Table 3 which you refer to in your text on page 9.

Table 3 was displayed on pages 13 and 14. We have edited the table and changed the layout and the order in which the information appear to be more readable. 

Thank you for your comment, we have added captions. 

8. Please review your reference list to ensure that it is complete and correct. If you have cited papers that have been retracted, please include the rationale for doing so in the manuscript text or remove these references and replace them with relevant current references. Any changes to the reference list should be mentioned in the rebuttal letter that accompanies your revised manuscript. If you need to cite a retracted article, indicate the article’s retracted status in the References list and also include a citation and full reference for the retraction notice.

Thank you, we followed your suggestion. 

Reviewers' comments:

• Reviewer #1: 

1. Giving information about urban and rural variations observed in HIV prevalence and other sexual behaviours indicators will be helpful for programme implementation by The Malawi National AIDS Commission, Malawi.

Thank you for this comment. We propose to modify the introduction section by adding a short sentence explaining that the prevalence vary according to gender, age, socio-economic characteristics, and geographic location in “introduction” : “In Malawi, where HIV prevalence remains one of the highest in the world, with variation in gender (10.8% for women; 6.4% for men), age, socio-economic characteristics and geographic location (14.6% for urban areas; 7.4% for rural areas), the rate is five times higher among adult FSWs (62.7%) [10–12]”. 

2. In general, the 15-24 age group HIV positive case is considered as a proxy for incidence since the study focussed only on younger FSWs. Removing the known HIV positives from databases and calculating the incidence % may be helpful to the programme.

Thanks for this thought-provoking comment. We should first clarify that the study did not focus only on younger FSWs, almost 75% were ≥20 years of age. Since the study was cross-sectional rather than cohort, we cannot report on incidence, but we agree that the rising prevalence with age gives a proxy. This proxy is difficult to interpret due to small sample sizes per age group and the uncertainty about how it relates to time in sex work and to other factors. 

• Reviewer #2: 

General Comments: This paper estimated HIV prevalence, HIV Cascade of Care, and STIs Prevalence among female sex workers (FSWs) in Malawi and by using 363 RDS samples. Moreover, this paper also examined FSWs' risk behaviors and sexual violence experience. Authors used Respondent-Driven Sampling (RDS) to recruit participants for a cross-sectional survey, which is appropriate given the ability of RDS to generate statistically unbiased estimates under certain conditions. Authors did clearly present methods of data collection, RDS estimators they used and sensitivity analysis they conducted. However, results of this paper, including tables, figures and interpretations, are somewhat flawed.

Thank you for your comment. We took heed that you observed the results in the paper were flawed and made the following changes described below. 

Major Comments:

1. Statistical tests were not mentioned in methods section or presented in tables/figures. However, authors used p-values in results section and discussion section.

Indeed, this was an accidental omission on our part. We added an explanation in the methods section (statistical methods): 

Difference in RDS estimates across sites were assessed using chi square test and the p-values presented. To examine the difference between adolescents and adults of some key indicators, we conducted logistic regression. These models excluded the seeds, weighted the data by the inverse of reported network size and obtained estimates after adjusting for site. 

2. Although author provided sub-headers in results section, which may help readers to easily understand, contents under each sub-header are still disorganized. Many results cannot be traced back to or supported by tables and figures. For instance, odds ratios that author mentioned in Line 155 and Line 156 are not found in tables or figures; VL and CD4 in Line 176 are not found in tables or figures. Results section should be more concise. Authors are highly recommended to keep comprehensive information in tables or figures and only interpret main findings. Authors may also need to tweak their interpretations, such as accurately interpreting odds ratios and avoiding using rates because of the study design.

The result section was shortened and reorganized as follow: 

- All results are now presented in the tables, this will ease the reading as results will be more easily traced back in tables.

- The paragraph on adolescents was moved to the end of the result section. Indeed, this specific analysis, carried on a sub-group of the population was not the main result of the paper. We now present the results, starting with those pertaining to the main study population and then progress through to results pertaining to subgroups.

- The CD4 result in line 176 was referring to a subgroup of individuals who did not know their HIV status. This was 10 in total and was not presented in the table; we have decided to drop this statement in order to avoid confusion.

- The VL result given in the text and not in the table was deleted (“including 2 who may have already been on treatment (VL<1000 copies/mL)”.

- We also added a specific table (table 4 at the end of the results section) to present the results of the logistic regression comparing the differences in some key indicators between adolescents and adults. 

- Table 2 and 3 were reorganized. 

3. Most of percentages or proportions in results section and discussion section are somewhat confusing. Authors may need to clearly present numerators and denominators for all percentages/proportions.

We have taken your comment on board and re-arranged the layout of the tables to follow the order in which the information appears in the text as much as possible. We grouped the key indicators under those associated to the whole study population under “Analysis based on study population” and the ones from analysis only conducted on a subgroup of the population under “Analysis based on a subgroup of study participants”. Therefore, results pertaining to subgroup analysis will not appear in the same order as the text in the table. 

We think it is worthwhile to present the denominators, but unnecessary to present the numerators as we are presenting the more informative proportions and 95% confidence interval from the RDS analysis. 

Further to reduce repeating the denominators on every line and given they are very similar; we include them at the top of each subsection of the table with either the actual denominator or the range in the denominator for the indicators listed. 

4. Table 2 and Table 3 are difficult to read. In the last column of Table 2, unexplained values do exist. Some values are confusing because authors did not present stratum-specific counts, numerators, and denominators. For instance, what does the number of being aware of HIV-positive status in Table 3 truly represent?

We re-arranged both table 2 and 3. In table 2, as the denominator was always the same for each variable in the different sites, we deleted the column “N” and placed it on top on the table. The mistakenly placed values were also deleted. In table 3, the unweighted results overall were deleted to simplify the table and all additional changes are mentioned above

Minor Comments:

1. In Line 37, what is the reference for the rate among adult FSWs to be five times higher? 

Thanks to your comment, we realized that there was a mistake in one of the references. We wanted to cite the Malawi Biological and Behavioural Surveillance Survey Report from 2014. This was changed in the manuscript. 

2. In Line 169, “95% CI” may be appropriate. 

We have done this change.

3. In Figure 1, authors are highly recommended to present overall HIV prevalence by age and site-specific HIV prevalence by age in single one plot. 

Figure 1 was modified accordingly to your comment. 

4. Figure 2 looks unnecessary. 

Ok we have deleted it. 

5. In Line 222, two percentages have completely different denominators, thus they should not be directly compared. 

Yes, you are right, we have reformulated the sentence to explain our idea better “Also, underreporting and recall bias are likely to affect estimates. For example, while only 48% of FSWs reported ever having experienced SV, the fact that so many of these (56%) reported it in the last month suggest the true lifetime experience of SV may be far higher”. 

6. In Line 272, authors said site specific RDS-estimates fulfill theoretical RDS assumptions. However, bottleneck plots in supplementary document may indicate preferential recruitment patterns and homophily. For instance, FSWs with a history of being enrolled in MSF activities were oversampled. Is it possible that FSWs with a history of being enrolled in MSF activities had higher HIV awareness and better treatment compliance and viral suppression comparing to FSWs without a history? 

Indeed, we recognize that FSWs enrolled in MSF activities may have been oversampled and that there may be homophily within the networks. However, we are unable to state whether this relates to a true underrepresented FSW population outside the networks identified through the seeds, or to a high coverage of the at-risk FSW population by MSF activities. We already discussed this in the “results and discussion” section: “Women receiving services from MSF-supported facilities were less likely to be virally suppressed (92.4% vs. 71.9%, p=0.008), suggesting that the peer-led approach may be identifying those FSWs with the greatest need for support”. 

• Reviewer #3: 

This paper describes an RDS study of a population of FSW in Malawi. These women are very vulnerable, and the inclusion of adolescent participants makes this a particularly important dataset.

I was impressed with the thoughtfulness and care that the authors demonstrated with this study. The inclusion of diagnostic plots as well as a frank discussion of convergence and bottleneck issues was quite refreshing. Many papers simply brush these issues under the rug.

I am comfortable recommending the paper for publication as is. Below are some optional comments for improvement.

Well noted, thank you. 

* I am not an expert on appropriate nomenclature; however, the authors may wish to review the application of the category “sex worker” to minors. The term implies a level of consent that may not be applicable to a trafficked 14-year-old.

We propose to change the nomenclature to “adolescent selling sex”. 

* Diagnostic results are presented in the discussion section, whereas it may be more appropriate to put this in the results section.

If you do not mind, we would prefer to leave them in the discussion section as they would require an entire section in the results to be presented. 

* It is probably worth noting that bottlenecks are to be expected between sites as they are geographically distinct and RDS chains cannot cross from one site to another. Even if the underlying network is well connected between sites, the RDS assumptions are violated because the chains cannot cross between them.

Yes, this is correct, and this is the reason why we opted for presenting the results by site specific indicators. We commented that in “statistical methods” (“Due to differences noted across sites during diagnostic checks, (differences in recruitment characteristics, varying chain lengths, bottlenecks), site-specific estimates were presented in addition to overall estimates) as well as at the end of “results and discussion” (“Recruitment occurred in three geographically distinct sites, and though site specific RDS-estimates fulfill theoretical RDS assumptions, district-level RDS-estimates may potentially be biased because we assume that participants from across this district are well-connected and that social networks may be linked (we did not collect sufficient information on the extent of an individual’s network between sites)”. 

* In addition to citing Stata and R, I recommend citing the RDS specific Stata and R packages used.

Ok we have added these 2 references. 

* Line 78: typo “two oded” 

Thank you we corrected this typo.

---

## [Decision Letter · Decision Letter 1]

13 Dec 2022

HIV, sexual violence, and termination of pregnancy among adolescent and adult female sex workers in Malawi: a respondent-driven sampling study

PONE-D-22-01162R1

Dear Dr. BOSSARD,

We’re pleased to inform you that your manuscript has been judged scientifically suitable for publication and will be formally accepted for publication once it meets all outstanding technical requirements.

Kind regards,

Henry F. Raymond

Academic Editor

PLOS ONE

Additional Editor Comments (optional):

Reviewers' comments:

Reviewer's Responses to Questions

**Comments to the Author**

1. If the authors have adequately addressed your comments raised in a previous round of review and you feel that this manuscript is now acceptable for publication, you may indicate that here to bypass the “Comments to the Author” section, enter your conflict of interest statement in the “Confidential to Editor” section, and submit your "Accept" recommendation.

Reviewer #2: All comments have been addressed

Reviewer #3: All comments have been addressed

2. Is the manuscript technically sound, and do the data support the conclusions?

Reviewer #2: Yes

Reviewer #3: Yes

3. Has the statistical analysis been performed appropriately and rigorously? 

Reviewer #2: Yes

Reviewer #3: Yes

4. Have the authors made all data underlying the findings in their manuscript fully available?

Reviewer #2: Yes

Reviewer #3: No

5. Is the manuscript presented in an intelligible fashion and written in standard English?

Reviewer #2: Yes

Reviewer #3: Yes

6. Review Comments to the Author

Reviewer #2: Thanks for your response to my previous comments and addressing them. I do not have any further comments.

Reviewer #3: Thank you to the authors for their work. I recommend this be accepted. Apparently this text box requires 100 characters.

7. PLOS authors have the option to publish the peer review history of their article (what does this mean?). If published, this will include your full peer review and any attached files.

Reviewer #2: No

Reviewer #3: No

---

## [Editor Report · Acceptance letter]

21 Dec 2022

PONE-D-22-01162R1 

HIV, sexual violence, and termination of pregnancy among adolescent and adult female sex workers in Malawi: a respondent-driven sampling study 

Dear Dr. Bossard:

I'm pleased to inform you that your manuscript has been deemed suitable for publication in PLOS ONE. Congratulations! Your manuscript is now with our production department. 

Kind regards, 

on behalf of

Dr. Henry F. Raymond 

Academic Editor

PLOS ONE